# Intelligent Grimm - Open-ended Visual Storytelling via Latent Diffusion Models

## Abstract

Benefiting from the impressive diffusion models, conditional generative models have exhibited exceptional capabilities in various generation tasks, for example, image or short video generation based on text description. In this work, we focus on the task of generating a series of coherent images based on a given storyline, denoted as *open-ended visual storytelling*. We make the following three contributions: (i) to fulfill the task of visual storytelling, we introduce two modules into a pre-trained stable diffusion model, and construct an auto-regressive image generator, termed as **StoryGen**, that enables to generate the current frame by conditioning on a text prompt and preceding frame; (ii) to train our proposed model, we collect paired image and text samples by sourcing from various online sources, such as videos, E-books, and establish a data processing pipeline for constructing a diverse dataset, named **StorySalon**, with a far larger vocabulary than existing animation-specific datasets; (iii) we adopt a three-stage curriculum training strategy, that enables style transfer, visual context conditioning, and human feedback alignment, respectively. Quantitative experiments and human evaluation have validated the superiority of our proposed model, in terms of image quality, style consistency, content consistency, and visual-language alignment. We will make the code, model, and dataset publicly available to the research community.

## 1 Introduction

*"Mirror, mirror, here I stand! Who is the fairest in the land?"*

*—- Grimms' Fairy Tales*

This paper explores an exciting yet challenging task of *visual storytelling*, with the goal of training a model that can effectively capture the relationship between visual elements in images and their corresponding language descriptions, to generate a sequence of images that tell a visual coherent story. The ultimate goal is to generate a sequence of images that tell a coherent story, provided in the form of natural language. The outcome of this task has significant potential for education, providing children with an engaging and interactive way to learn complex visual concepts, and develop imagination, creativity, emotional intelligence, and language skills, as evidenced by the research in psychology [4, 41].

In the recent literature, there has been significant progress in image generation, particularly with the guidance of text, such as stable diffusion [35], DALL·E [33] and Imagen [9]. However, these models are not sufficient for visual storytelling for two reasons: (i) existing models generate images independently without considering the context of previous frames or the overall narrative, resulting in visual inconsistencies and a lack of coherence in the visual story; (ii) generating images by only conditioning on text can lead to ambiguities or require unnecessarily long descriptions, particularly

Submitted to 37th Conference on Neural Information Processing Systems (NeurIPS 2023). Do not distribute.

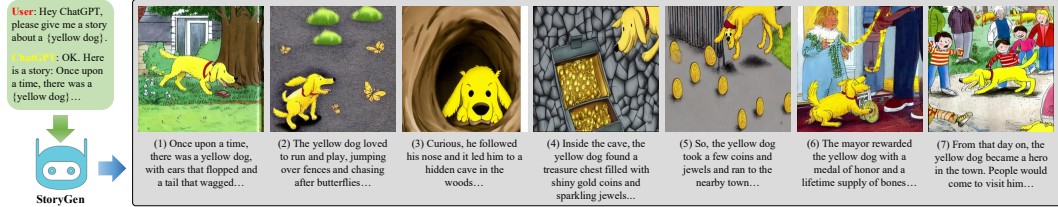

Figure 1: An illustration of open-ended visual storytelling. In practise, one user can prompt a large larnguage model, for example, ChatGPT, to generate a unique and engaging story, which is then fed into our proposed **StoryGen** model, to generate a sequence of images that are not only aligning to the given storyline, but also coherent. We recommend the reader to zoom in and read the story.

when dealing with subtle differences, for example, distinguishing between animals from the same category but different breeds. To address the limitations, we introduce a novel auto-regressive architecture, termed as **StoryGen**, that builds upon pre-trained stable diffusion model, with two extra modules serving for style transfer and visual context conditioning. At inference time, StoryGen takes the preceding frames and text prompts as conditions to synthesize the current frame, *i.e.*, iteratively creating visual sequences that are not only aligning to language description, but also coherent.

In practise, visual storytelling is faced with significant challenges due to the lack of high-quality image-text data. Most existing works are limited to train on a few specific animations, for example, StoryGAN [17], StoryDALL-E [22] and AR-LDM [28], resulting in a small and restricted vocabulary and characters. To overcome this limitation, we have created a dataset called **StorySalon**, that features a rich source of coherent images and stories, primarily comprising children's storybooks collected from three different sources: videos, E-books, and synthesized sample data from our StoryGen model with human verification. As a result, our dataset includes a diverse vocabulary with different characters, storylines, and artistic styles.

We follow a three-stage training procedure: *Firstly*, we insert a LoRA-like architecture into text conditioning module in stable diffusion model, *i.e.*, on top of the image-to-text cross-attention, in order to adapt the pre-trained diffusion model on our collected dataset; *Secondly*, we introduce a visual context module that enables the generation process to condition on preceding image; *Lastly*, we finetune the model with data after human verification, *i.e.*, leveraging the feedback to further align the model with human preference. As a result, the scale and diversity of our collected dataset enable the model to acquire the ability of open-vocabulary visual storytelling, by that we mean, our model can generate new image sequences that are not limited to pre-defined storylines, characters, or scenes. For example, we can prompt a large language model to create unique and engaging stories, then feed into StoryGen for generation, as shown in Figure 1.

To summarise, we make the following contributions in this paper: (i) we propose the task of *open-ended visual storytelling*, that involves generating engaging image sequence that is aligning to a given storyline, for example, written by a large language model; (ii) we develop a novel architecture based on stable diffusion, termed as StoryGen, which can generate image sequences in an auto-regressive manner, taking both preceding image and text prompt of current frame as condition; (iii) we initiate a data collection pipeline and collect a large-scale, diverse datasets of storybooks, from online videos, E-books and synthesized samples, including paired image-text samples of a diverse vocabulary with different characters, storylines, and artistic styles; (iv) we adopt a three-stage curriculum training strategy, that enables style transfer, visual context conditioning, and human feedback alignment, respectively. Experimentally, we conduct both quantitative comparison and human evaluation, showing that the outputs from ou proposed model are more preferred, in terms of image quality, style consistency, and image coherence.

## 2   Related Works

**Diffusion Models** learn to model a data distribution via iterative denoising and are trained with denoising score matching. Drawing from the principles of Langevin dynamics and physical diffusion processes, diffusion models have undergone refinement through a multitude of works [39, 27, 45]. Notably, DDPM [10] has demonstrated improved performance over other generative models, while DDIM [40] has significantly boosted generation efficiency. In view of their superior generative

capabilities, diffusion models have found extensive utility in various downstream applications besides image generation, such as video generation [46, 5, 12, 9, 38], image manipulation [2, 24, 14, 7], grounded generation [18], 3D texturing [34], and image inpainting [47, 26, 19, 1].

**Text-to-image Generation** involves the creation of images from textual descriptions. The task has been tackled using various generative models, with Generative Adversarial Networks (GANs) [6] being the first widely-used model. Several GAN-based architectures such as StackGAN [50], StackGAN++ [51], and AttnGAN [48] have achieved notable success in this area. Additionally, pre-trained auto-regressive transformers [43] such as DALL·E [33] have demonstrated the ability to generate high-quality images in response to textual prompts. Recently, diffusion models have emerged as a popular approach to text-to-image generation. New images can be sampled under text guidance from the data distribution learned by diffusion models with iterative denoising process. DALL·E 2 [32] leverages CLIP [29] features to achieve well-aligned text-image latent space, while Imagen [37] relies on large language models like T5 [31] to encode text. Stable Diffusion (or Latent Diffusion) Model [35] performs diffusion process in latent space, and it can generate impressive images after pre-training on a large-scale text-image datasets.

**Story Synthesis** is first introduced as the task of story visualization (SV) by StoryGAN [17], which presents a GAN-based framework and the inaugural dataset named Pororo, derived from cartoons. Subsequently, some other works also follow the GAN-based framework, such as DUCO-StoryGAN [21] and VLC-StoryGAN [20]. In the case of word-level SV [16], more emphasis is placed on the representation of text, whereas VP-CSV [3] employs VQ-VAE [42] and a transformer-based language model to conserve character appearance and enhance visual quality. StoryDALL·E [22] extends the story visualization task to story continuation with the initial image given and recommends using a pre-trained DALL·E generative model [33] to produce coherent images. AR-LDM [28] introduces an auto-regressive latent diffusion model that can generate highly realistic images, but with only a limited vocabulary. NUWA-XL [49] is a concurrent work that exploits hierarchical diffusion model to synthesize long videos, with the keyframes generated first, followed by frame interpolation.

Existing models are mostly developed on specific scenes, which limits their ability for generating image sequences for diverse stories. In this paper, we target for more ambitious applications, to develop an open-ended visual storytelling model, that can digest stories of arbitrary length and diverse topics, and synthesize a sequence of coherent images in terms of both style and semantic.

# 3 Method

To be self-contained, we first present a brief overview to diffusion model in Section 3.1; then, we detail our proposed model for storybook generation in Section 3.2, starting from problem formulation, then architecture details, and lastly on training details.

## 3.1 Preliminaries on Diffusion Models

Diffusion models are a type of generative models that undergo a denoising process, converting input noise into meaningful data samples. Diffusion models comprise a forward diffusion process that incorporates Gaussian noise into an image sample $\mathbf{x}_0$, accomplished via a Markovian process over $T$ steps. If we denote the noisy image at step $t$ as $\mathbf{x}_t$, the transition function $q(\mathbf{x}_t|\mathbf{x}_{t-1})$ connecting $\mathbf{x}_{t-1}$ and $\mathbf{x}_t$ can be expressed as follows:

$$q(\mathbf{x}_t|\mathbf{x}_{t-1}) = \mathcal{N}(\mathbf{x}_t; \sqrt{1-\beta_t}\mathbf{x}_{t-1}, \beta_t\mathbf{I}) \quad q(\mathbf{x}_{1:T}|\mathbf{x}_0) = \prod_{t=1}^{T} q(\mathbf{x}_t|\mathbf{x}_{t-1}) \tag{1}$$

where $\beta_t \in (0,1)$ is the variance schedule controlling the step size.

Using Gaussian distribution property and reparametrization, if we define $\alpha_t = 1 - \beta_t$ and $\bar{\alpha}_t = \prod_{i=1}^{t} \alpha_i$, we can write equation 1 as follows:

$$q(\mathbf{x}_t|\mathbf{x}_0) = \mathcal{N}(\mathbf{x}_t; \sqrt{\bar{\alpha}_t}\mathbf{x}_0, (1-\bar{\alpha}_t)\mathbf{I}) \tag{2}$$

Diffusion models also comprise a reverse diffusion process that learns to restore the initial image sample from noise. A UNet-based model [36] is utilized in the diffusion model to learn the reverse

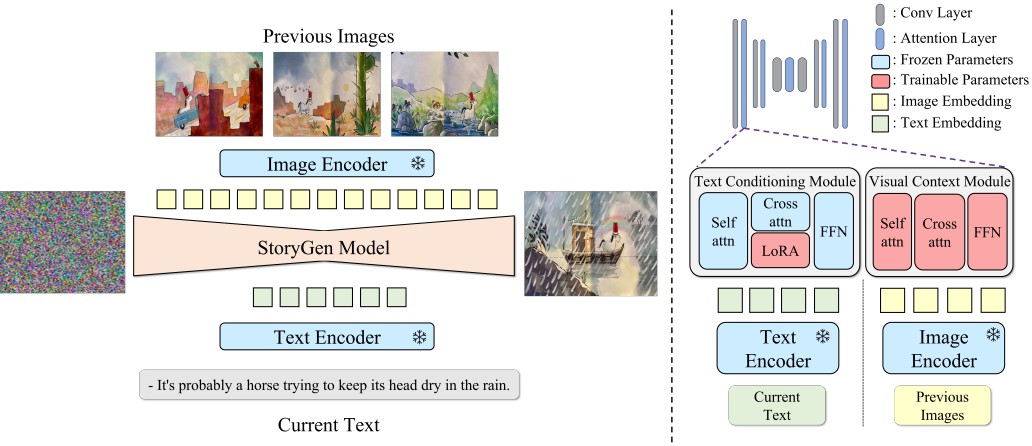

Figure 2: **Architecture Overview**. The **left** figure illustrates the complete procedure of *visual storytelling*. Our StoryGen model utilizes contextual information from previous frame and the text description at current step, to generate an image. The **right** figure displays the structure of our proposed modules, (i) style transfer module that is inserted into the text-conditioning module, with a LoRA-like achitecture; (ii) visual context module that enables the model to also condition on the features from preceding image for generation.

diffusion process $p_\theta$. The process $p_\theta$ can be expressed using the following equation.

$$p_\theta(\mathbf{x}_{0:T}) = p(\mathbf{x}_T) \prod_{t=1}^{T} p_\theta(\mathbf{x}_{t-1}|\mathbf{x}_t) \quad p_\theta(\mathbf{x}_{t-1}|\mathbf{x}_t) = \mathcal{N}(\mathbf{x}_{t-1}; \boldsymbol{\mu}_\theta(\mathbf{x}_t, t), \boldsymbol{\Sigma}_\theta(\mathbf{x}_t, t)) \quad (3)$$

where $\boldsymbol{\mu}_\theta$ is the predicted Gaussian distribution mean value.

As we compute the loss function by taking the mean absolute error of the noise term $\boldsymbol{\epsilon}_\theta$ into account, we can express the mean value $\boldsymbol{\mu}_\theta$ in terms of the noise term $\boldsymbol{\epsilon}_\theta$ as follows:

$$\boldsymbol{\mu}_\theta(\mathbf{x}_t, t) = \frac{1}{\sqrt{\alpha_t}} \left( \mathbf{x}_t - \frac{1-\alpha_t}{\sqrt{1-\bar{\alpha}_t}} \boldsymbol{\epsilon}_\theta(\mathbf{x}_t, t) \right) \quad (4)$$

Therefore, the objective can be written as:

$$\mathcal{L}_t = \mathbb{E}_{t \sim [1,T], \mathbf{x}_0, \boldsymbol{\epsilon}_t} \left[ \|\boldsymbol{\epsilon}_t - \boldsymbol{\epsilon}_\theta(\mathbf{x}_t, t)\|^2 \right] \quad (5)$$

### 3.2 StoryGen Model

In this section, we start by defining the problem for visual storytelling, then we introduce the core components in our proposed architecture, namely, Style Transfer Module and Visual Context Module, lastly, we present details for training the model with a curriculum learning regime.

#### 3.2.1 Problem Formulation

In visual storytelling, our goal is to generate a sequence of coherent and consistent images that correspond to a given story in the form of natural language. To achieve this, we propose an auto-regressive image generation model, called **StoryGen**, that generates the current frame $\mathcal{I}_k$ by conditioning on both current text description $\mathcal{T}_k$ and the previous frame $\mathcal{I}_{k-1}$, as illustrated in Figure 2. The model is formulated as follows:

$$\{\hat{\mathcal{I}}_1, \hat{\mathcal{I}}_2, \dots, \hat{\mathcal{I}}_L\} = \Phi_{\text{StoryGen}}(\{\mathcal{T}_1, \mathcal{T}_2, \dots, \mathcal{T}_L\}; \Theta),$$

$$= \psi_{\text{SDM}}(\hat{\mathcal{I}}_1|\mathcal{T}_1; \theta) \dots \psi_{\text{SDM}}(\hat{\mathcal{I}}_k|\hat{\mathcal{I}}_{k-1}, \mathcal{T}_k; \theta) \dots \psi_{\text{SDM}}(\hat{\mathcal{I}}_L|\hat{\mathcal{I}}_{L-1}, \mathcal{T}_L; \theta)$$

Here, $\{\mathcal{T}_1, \mathcal{T}_2, \dots, \mathcal{T}_L\}$ represents the given story in text sentences, $\{\hat{\mathcal{I}}_1, \hat{\mathcal{I}}_2, \dots, \hat{\mathcal{I}}_L | \hat{\mathcal{I}}_i \in \mathbb{R}^{H \times W \times 3}\}$ denotes the generated storybook, $H, W$ refer to the width and height, respectively. $\psi_{\text{SDM}}(\cdot)$ refers

to a semi-frozen stable diffusion model (SDM), with a small number of newly-added trainable
parameters ($\theta$). It takes randomly sampled gaussian noise, text description and preceding image as
input, and generate coherent image sequence that align with the story's narrative. In the following
sections, we present the architecture detail for one-step generation conditioned on text and image.

### 3.2.2 Architecture Details

Generally speaking, our model is built upon the foundation of a pre-trained stable diffusion
model (SDM), that has been pre-trained on large number of paired image-caption samples, to
gradually transform the noisy latent into an image. To tackle the problem of open-ended storytelling,
we introduce two computational modules, namely, Style Transfer Module, and Visual Context Mod-
ule, that enables the model to condition on not only text descriptions, but also the preceding RGB
image, as shown in Figure 2. Formally, we can express the generation procedure as:

$$\mathcal{I}_k = \psi_{\text{SDM}}(\hat{\mathcal{I}}_k | \hat{\mathcal{I}}_{k-1}, \mathcal{T}_k) = \psi_{\text{SDM}}(\mathbf{x}, \phi_{\text{text}}(\mathcal{T}_k), \phi_{\text{vis}}(\hat{\mathcal{I}}_{k-1}))$$

where $\mathbf{x}$, $\phi_{\text{text}}(\cdot)$ and $\phi_{\text{vis}}(\cdot)$ denote the noisy latent, encoded text description and preceding image.

**Style Transfer Module.** To steer a pre-trained stable diffusion model towards the style of children's
storybooks, we propose to insert a lightweight, LoRA-like [13] architecture into the text conditioning
module, effectively acting as style transfer. This can be expressed as:

$$Q = W^Q \cdot \mathbf{x} + \Delta Q, \ K = W^K \cdot \mathcal{C}_k^{\text{text}} + \Delta K, \ V = W^V \cdot \mathcal{C}_k^{\text{text}} + \Delta V, \ \text{where} \ \mathcal{C}_k^{\text{text}} = \phi_{\text{text}}(\mathcal{T}_k)$$

$W^Q, W^K$ and $W^V$ denote the projection matrices, adopted from the text conditioning module in
pre-trained stable diffusion model. $\phi_{\text{text}}(\cdot)$ and $\mathcal{C}_k^{\text{text}}$ refer to the pre-trained CLIP text encoder and
extracted text embedding respectively. $\Delta Q, \Delta K$ and $\Delta V$ are calculated by a learnable projection of
$\mathbf{x}, \mathcal{C}_k^{\text{text}}$ and $\mathcal{C}_k^{\text{text}}$, respectively, resembling LoRA operations.

**Visual Context Module.** In order to generate visually coherent images, we insert a visual context
module after the text conditioning, specifically, it is a transformer decoder comprising a self-attention
layer, a cross-attention layer, and a feed-forward network, where the cross-attention layer employs a
casual attention mechanism by using the noisy latent as query and the visual features of the previous
frame as key and value, which can be formally denoted as:

$$Q = W^Q \cdot \mathbf{x}, \quad K = W^K \cdot \mathcal{C}_k^{\text{vis}}, \quad V = W^V \cdot \mathcal{C}_k^{\text{vis}}, \quad \text{where} \ \mathcal{C}_k^{\text{vis}} = \phi_{\text{vis}}(\mathcal{I}_{k-1})$$

$W^Q, W^K$ and $W^V$ refer to three learnable projection matrices, $\phi_{\text{vis}}(\cdot)$ denotes the visual feature
extracted by a pre-trained CLIP visual encoder. It is worth noting that the visual context module can
also extend to multiple condition frames by concatenating their CLIP features as visual contexts.

**Training Objective.** At training stage, we randomly sample a triplet each time, *i.e.*, $\{\mathcal{I}_k, \mathcal{I}_{k-1}, \mathcal{T}_k\}$,
and the objective defined in Equation 5 can now be transformed into:

$$\mathcal{L}_t = \mathbb{E}_{t \sim [1,T], \mathbf{x}_0, \boldsymbol{\epsilon}_t, \mathcal{C}_k^{\text{vis}}, \mathcal{C}_k^{\text{text}}} \left[ \| \boldsymbol{\epsilon}_t - \boldsymbol{\epsilon}_\theta(\mathbf{x}_t, t, \mathcal{C}_k^{\text{vis}}, \mathcal{C}_k^{\text{text}}) \|^2 \right] \tag{6}$$

and as we adopt classifier-free guidance [11] in inference, the predicted noise can be expressed as:

$$\bar{\boldsymbol{\epsilon}}_\theta(\mathbf{x}_t, t, \mathcal{C}_k^{\text{vis}}, \mathcal{C}_k^{\text{text}}) = (w+1)\boldsymbol{\epsilon}_\theta(\mathbf{x}_t, t, \mathcal{C}_k^{\text{vis}}, \mathcal{C}_k^{\text{text}}) - w\boldsymbol{\epsilon}_\theta(\mathbf{x}_t, t) \tag{7}$$

where $w$ is the guidance scale.

### 3.2.3 Curriculum Learning

In this section, we describe the three-stage training strategy, that includes single-frame pre-training,
multiple-frame fine-tuning, and alignment with human feedback. This curriculum learning approach
enables the model to gradually learn from simple to complex tasks, ultimately improving its ability
to generate high-quality images that align with the given story narratives. Details for our proposed
curriculum learning are presented below.

**Single-frame Pre-training.** We start by training the style transfer module, which has been inserted
into the text conditioning module of a pre-trained stable diffusion model, in a single-frame manner. At
this pre-training stage, we do not introduce the visual context module, and freeze all other parameters
except for the LoRA-like plug-in. This training approach allows us to quickly adjust to the desired

visual style and character appearance in storybooks, while also maintaining the generation ability of the pre-trained stable diffusion model.

**Multiple-frame Fine-tuning.** Here, we fine-tune the visual context module while freezing other parameters of the model. Till this point, this allows the generation procedure to utilize information from either the text description or the preceding frame. To avoid over-fitting to the text descriptions, we adopt a technique inspired by BERT training [15], randomly dropping some words in the text with certain probability. The entire visual context module is fine-tuned during this stage.

**Fine-tuning with Human Feedback.** After multiple-frame fine-tuning, the model has developed basic storybook generation capabilities, to avoid from generating, potentially scary, toxic or biased content, we also propose to align the model with human preference. Specifically, we prompt ChatGPT to generate approximately 200 stories and use our model to synthesize images. After manually filtering around 100 high-quality storybooks from this corpus, we add them to the training set for further fine-tuning. As future work, we aim to add more books into this human feedback step.

**Inference.** With the three-stage training regime, we can streamline the entire inference process into a unified generation framework. As shown in the Figure 1, at inference time, we can prompt the ChatGPT to generate engaging, yet educational storylines, and synthesize the first image using a single-frame approach with only style transfer module involved; the previously synthesized frames, along with story description at current step, are treated as condition to generate image sequence in an auto-regressive manner. Experimentally, our proposed StoryGen is shown to generate images that align with the storyline, as well as maintaining consistency with previously generated frames.

# 4 Dataset Preparation

For training visual storytelling model, we collect a dataset called **StorySalon**, that contains approximately 2K storybooks and more than 30K well-aligned text-image pairs. This dataset is comprised of storybooks with potentially aligned text and image pairs, sourced from three different sources: video data, E-book data, and additional data from human feedback.

## 4.1 Image-text Data from Videos & E-books

Here, we elaborate the procedure of extracting paired image-text samples from YouTube videos and E-books (pdf and corresponding audios available).

**Visual Frame Extraction.** To begin with, we download a significant amount of videos and subtitles from YouTube, by querying keywords related to children story, for example, *storytime*. We then extract the keyframes from the videos, along with the corresponding subtitles and their timestamps. To remove duplicate frames, we extract ViT features for each frame using pre-trained DINO [23], for the image groups with high similarity score, we only keep one of them. Next, we use YOLOv7 [44] to segment and remove person frames and headshots, as they often correspond to the story-teller and are unrelated to the content of the storybook. Finally, we manually screen out frames that are entirely white or black. Similarly, we also acquire a number of electronic storybooks from the Internet, and extract images from E-book, except for those with extraneous information, for example, the authorship page. We acquire the corresponding text description with Whisper [30] from the audio file. For E-books that do not have corresponding audio files, we use OCR algorithms, to directly recognize the text on each page.

**Visual-Language Alignment.** Here, for each image, we acquire two types of text description, namely, story-level description, and visual description. This is based on our observation that there actually exists semantic gap between story narrative and descriptive text, for example, the same image can be well described as *"The cat is isolated by others, sitting alone in front of a village."* in story, or *"A black cat sits in front of a number of houses."* as visual description, therefore, directly finetuning stable diffusion model with story narrative maybe detrimental to its pre-trained text-image alignment.

In practise, to get story-level paired image-text samples, we align the subtitles with visual frames by using Dynamic Time Warping (DTW) algorithm [25]. To get visual descriptions, we use ChatCaptioner [52] to generate captions for each image, as shown in Figure 3. s At training time, this allows us to substitute the original story with more accurate and descriptive captions.

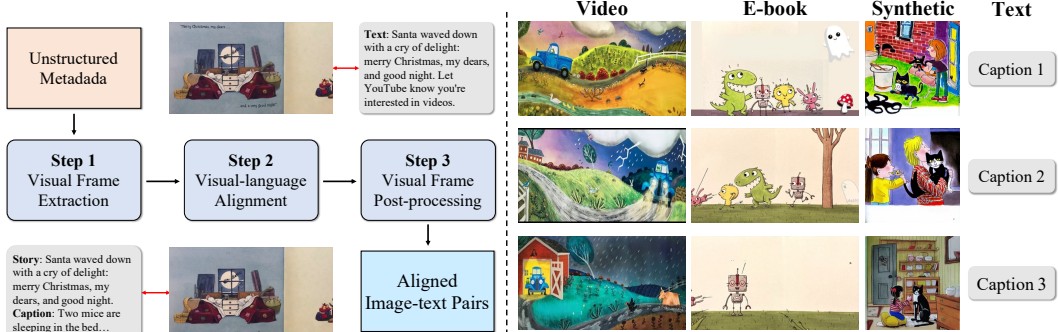

Figure 3: **Dataset Pipeline Overview**. The **left** figure provides an overview of the complete dataset collection pipeline. Unstructured metadata sourced from the Internet undergoes a series of steps including frame extraction, visual-language alignment and image inpainting, resulting in properly aligned image-text pairs. The **right** figure displays examples of video data, E-book data, and synthetic samples. The accompanying texts represent their corresponding textual content, respectively.

**Visual Frame Post-processing.** In practice, we discovered that books in the frames can potentially interfere with our image generation model by having story texts printed on them. To address this, we use an OCR detector to identify the text regions in the frames and an image inpainting model to fill in the text and headshot regions. This process results in more precise image-text pairs that can be fed into the diffusion model.

## 4.2 Additional Data from Human Feedback

As outlined in Section 3.2.3, we use the model (trained after two stages) to generate a set of new storybooks, and incorporate human feedback into the fine-tuning process. Following a rigorous manual review, we carefully select the best pieces, and add them into the training dataset. This allows us to continually improve the quality of our model and ensure that it produces engaging, yet educational storybooks, that align with human's preference.

## 4.3 Discussion

Given the diversity of our data sources and data types, the StorySalon dataset exhibits a significantly broader range of visual styles and character appearances over other animation-specific datasets. Moreover, our dataset surpasses others in terms of vocabulary coverage by a substantial margin. Notably, our texts seamlessly integrate narrative storylines and descriptive visual prompts, ensuring the preservation of text-image alignment while adapting to the art style of storybooks.

## 5 Experiment

In this section, we start by describing our experimental settings, then compare with other models from three different perspectives: style, quality and coherence with quantitative and subjective human evaluation. Additionally, we present the results of our ablation experiments to prove the effectiveness of our proposed training regime.

## 5.1 Training Settings

Our model is based on publicly released stable diffusion checkpoints, with a learning rate of $1 \times 10^{-5}$ and a batch size of 512. We begin with a single-frame pre-training stage, which involves 10,000 iterations on 8 NVIDIA RTX3090. In the multiple-frame fine-tuning stage, we fine-tune the model for 40,000 iterations using a single condition image. To improve the robustness of the training procedure, we apply a $10\% \sim 30\%$ words dropout with a probability to the texts in the current frames. We also fine-tune the model with human feedback on the training set with 100 manually generated storybooks in addition for 5,000 iterations. During inference, we utilize DDIM sampling and classifier-free guidance with a weight of 6.0.

| Model | FID ↓ | Alignment ↑ | Style ↑ | Content ↑ | Quality ↑ | Preference |
|---|---|---|---|---|---|---|
| GT | - | 4.22 | 4.68 | 4.34 | 4.32 | - |
| **StoryGen** | **120.01** | **4.02** | **3.82** | **3.67** | **3.53** | **70.42%** |
| Prompt-SDM | 167.21 | 3.45 | 2.25 | 2.64 | 3.21 | 16.11% |
| SDM | 184.01 | 3.28 | 2.56 | 2.57 | 3.49 | 13.47% |

Table 1: **Comparison result of human evaluation and FID.** GT stands for the ground truth from the training set. SDM denotes Stable Diffusion and Prompt-SDM denotes SDM with cartoon-style-directed prompts.

| Model | FID ↓ |
|---|---|
| with HF | **66.41** |
| without HF | 66.60 |
| Prompt-SDM | 101.23 |
| SDM | 115.43 |

Table 2: Ablation study on human feedback.

## 5.2 Quantitative Results

To evaluate the quality of our generated image sequence, we adopt the widely-used Fréchet Inception Distance (FID) score [8]. However, as there is no standardized metric for evaluating the consistency of images, we include human evaluation for comparison.

**Fréchet Inception Distance (FID).** We present a comparison of the FID scores between our model and other existing ones, including SDM and Propmt-SDM, which conditions on an additional cartoon-style-directed prompt *"A cartoon style image"*. Specifically, we calculate the FID scores between the distribution of the generated results from these models and the distribution of the our proposed **StorySalon** testset. As shown in Table 1, we evaluate the generated image sequences from 100 storylines, obtained by prompting ChatGPT. Our StoryGen model outperforms the original stable diffusion models (SDM) and Prompt-SDM by a large margin, demonstrating the effectiveness of our model in generating high-quality coherent images.

**Human Evaluation.** We conduct two types of human evaluation experiments to assess the quality of our generated storybooks. In the *first* experiment, we randomly select an equal number of groundtruth storybooks, the results of our StoryGen and the generation results of SDM and Propmt SDM. Participants are then invited to rate these four categories of storybooks on a score ranging from 1 to 5, taking into account text-image alignment, style consistency, content consistency, and image quality, higher scores indicate better samples. In the *second* experiment, we prompt ChatGPT to produce a number of storylines and use our StoryGen along with the two variations of stable diffusion to generate corresponding image sequences. Participants are asked to choose the preferred results of each storyline. To mitigate bias, participants are unaware of the type of storybooks they are evaluating during these two human evaluation experiments. In both experiments, we have invited approximately 30 participants in total.

Table 1 presents the results of our human evaluation. As can be seen, our model has shown significant performance improvement in its overall score compared to stable diffusion models, especially in terms of consistency and alignment, indicating that it can generate images that are highly consistent with the given text prompts and visual contexts, thus better exploiting the contextual information.

**Ablation Studies.** Our study compares the performance of our model with and without fine-tuning using human feedback. We evaluate the FID score of between our generated image sequence and the test set of our StorySalon dataset. The results demonstrate that fine-tuning with human feedback slightly improves performance, we conjecture that this could be attributed to the fact that the number of human-verified samples are limited, due to a result of resource limitation. In future work, we intend to address this quantity constrain by continually augmenting the dataset with new samples and closely monitoring the progressive advancements.

## 5.3 Qualitative Results

In Figure 4, we present the visualization results, showing that our model can generate storybooks with a broad vocabulary, while maintaining coherence and consistency throughout the narrative. The generated images successfully maintain the consistency of the artistic style and character appearance, whereas the results from SDM and Prompt-SDM fail to do so. Moreover, style of the generated results from SDM's are also incongruent with the requirements of visual storytelling for childrens.

## 6 Conclusion

In this paper, we consider the exciting yet challenging task known as *open-ended visual storytelling*, which involves generating a sequence of consistent images that tell a coherent visual story based

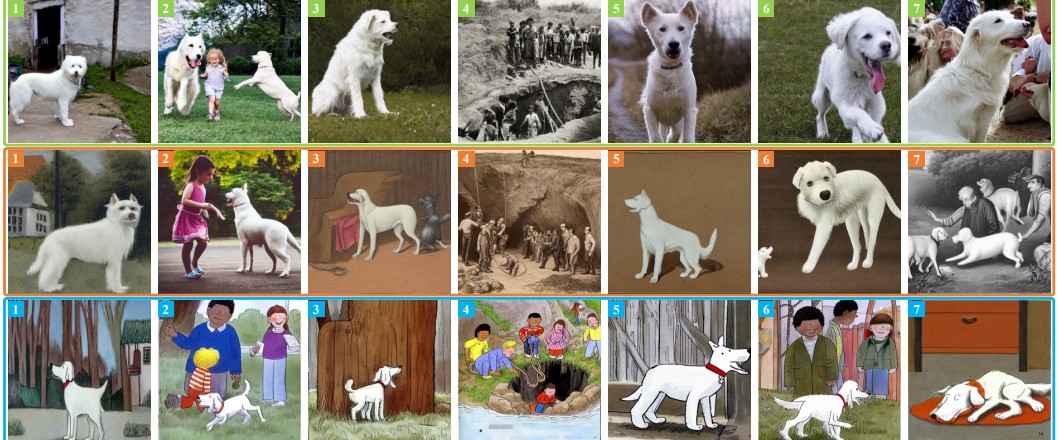

(a) A story of a {white dog}: (1) Once upon a time, in a small village, there lived a white dog. It had pure white fur that sparkled in the sunlight. (2) The white dog was loved by all the villagers. Children would play with it, and adults would often walk it around the village. (3) One day, while the white dog was taking a walk, it heard a cry for help. It followed the sound and found a young boy who had fallen into a deep pit. The white dog quickly sprang into action and started barking loudly to get the attention of the villagers. (4) Within minutes, a group of villagers gathered around the pit and lowered a rope to rescue the boy. (5) From that day onwards, the white dog was regarded as a hero in the village. (6) Eventually, the white dog became old. But even in its old age, the white dog would still wag its tail whenever it saw a child or heard someone call its name. (7) The white dog passed away peacefully, surrounded by the love and affection of the entire village.

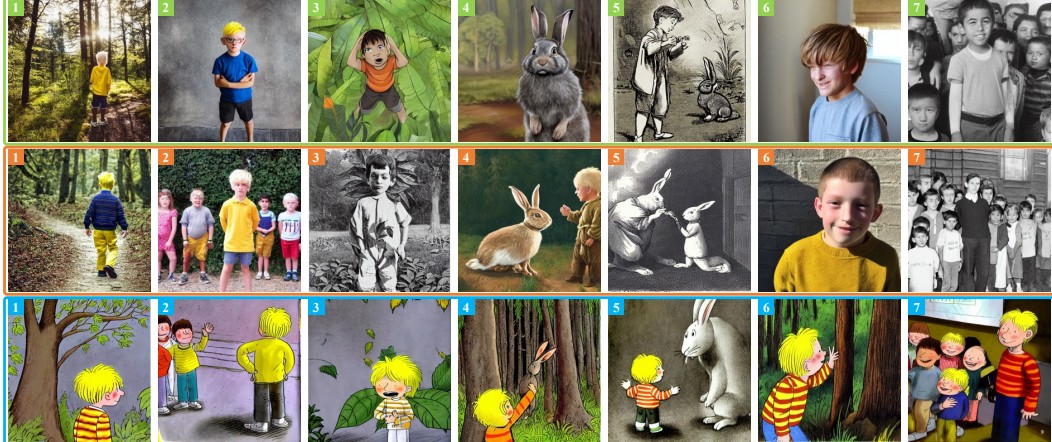

(b) A story of a {boy with yellow hair}: (1) Once upon a time, there was a boy with yellow hair as bright as the sun. (2) The boy loved his yellow hair, even though other children teased him for looking different. (3) One day, as the boy was walking through the forest, he stumbled upon a magical vase. When he touched the vase, he suddenly shrunk down to the size of a bug. (4) Scared and confused, the boy tried to find his way back to his normal size. He wandered through the forest, until he came across a wise old rabbit. (5) The rabbit taught him a special spell to grow back to his normal size. The boy was grateful for the rabbit's kindness. (6) As the boy regained his size, he realized that his yellow hair had grown even brighter because of the magical vase. He ran back to his village, excited to show off his hair. (7) But when the children saw his hair, they no longer wanted to tease him. They were amazed by the boy's bold and beautiful yellow hair. From that day forward he had grown to understand that being different is what makes us special, and he embraced his uniqueness with pride.

Figure 4: **Qualitative Comparison with other baselines**. The images in green, orange and blue boxes are generated SDM, Prompt-SDM and StoryGen respectively. Our results have superior style and content consistency, text-image alignment, and image quality.

on the given storyline. Our proposed **StoryGen** architecture can take input from the preceding frame along with the text prompt to generate the current frame in an auto-regressive manner. A three-stage curriculum training strategy has been introduced for effective training and alignment with human preference. Due to the limitations of dataset in previous works, we have also collected a large-scale, diverse dataset named **StorySalon** that includes paired image-text samples sourced from storybook data from videos, e-books and synthesized samples. The StorySalon dataset possesses a diverse vocabulary of storyline, character appearances and artistic styles. While comparing with stable diffusion models with quantitative experiment and human evaluation, our proposed model substantially outperform existing models, from the perspective of image quality, style consistency, content consistency, and image-language alignment.

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
