# OpenReview forum: "Intelligent Grimm - Open-ended Visual Storytelling via Latent Diffusion Models"
_NeurIPS.cc/2023/Conference — Submitted to NeurIPS 2023_

### Official Review · Reviewer_zaiF · 2023-06-25

**Soundness:** 2 fair
**Presentation:** 3 good
**Contribution:** 2 fair
**Rating:** 4
**Confidence:** 4

**Summary:**

The authors aim to generate a series of coherent images given a series of text prompts resembling a visual storybook. To do so, the authors focus on two fronts: (1) leveraging the Stable Diffusion model to generate the series of images and (2) generating a diverse dataset used to train the model on a range of styles. For generating a set of coherent images, the authors condition the Stable Diffusion on both the text prompt and a set of previously generated frames, both encoded using a frozen CLIP encoder. The text conditioning is passed to the U-Net layers via the standard cross-attention module and LoRA. To insert the image conditioning, the authors introduce a Visual Context Model resembling the standard text conditioning module. The diffusion model is then partially trained to generate images that are both consistent with the text prompt and previously generated frames. To attain images ranging in style, the authors construct a new dataset, named StorySalon, consisting of Youtube videos and E-Books. These raw storybooks are filtered and re-captioned to better align with the visual content of the storybook. Qualitative results demonstrate the ability to generate new storybooks on prompts generated by ChatGPT while quantitative results demonstrate improvements over simple baselines.

**Strengths:**

- The authors focus on an important task of generating a series of coherent images that follow a given text prompt. This could potentially be useful beyond a simple storybook creation, e.g., for video generation.
- The Visual Context Module as a means for injecting image-level details to the denoising network is simple and intuitive and could be useful in other tasks. For example, in image editing where the desired edit cannot be easily described using language.
- The visual results generated by StoryGen are impressive in comparison to the evaluated baselines.


**Weaknesses:**

**Dataset Creation:**
- I have some reservations regarding the use of the term Human Feedback in Section 3.2.3 and Section 4.2. While ChatGPT was fine-tuned to align with human preference, I believe that using ChatGPT for generating additional prompts should not be considered Human Feedback. While this is an intriguing approach, I believe replacing Human with LLM is more reflective of what is actually done here.
- Regarding the ablation study performed by the authors, it seems that “Human Feedback” leads to a quite negligible decrease in FID and I am therefore uncertain if this really contributes to the curriculum learning scheme. The authors mention that more stories can be added, but I would have expected to see a bigger improvement if this stage is truly important.

**Evaluation:**
- There are numerous essential evaluations that are missing from the current submission. Among these, the most important is a thorough evaluation and comparison of StoryGAN, Story-DALL-E, and AR-LDM. All three have publicly available code so an evaluation is needed to understand the improvement realized by StoryGen.
- I am not sure that FID is a particularly interesting metric here since all evaluated methods in Table 1 use Stable Diffusion to generate the images. Moreover, I do not believe that FID is a good metric when trying to measure how much the image captures a given style, as is the goal here. Maybe a CLIP-based metric using a prompt depicting a style would be more appropriate here?
- There are numerous ablation studies that I believe are required to understand the contribution of both the proposed architecture and dataset.
    - Architecture:
        - An ablation study on the Visual Context Module and whether a simpler conditioning is possible (see my detailed question below).
        - Was an ablation study performed on the BERT-like masking during the multi-frame fine-tuning?
    - Dataset
        - The impact of the visual-language alignment stage in preparing the dataset. The authors state that directly fine-tuning on the story narrative may be detrimental, but do not validate this claim.
- Some additional evaluations could help validate the effectiveness of the method.
    - First, the authors claim that the method can be used to generate stories of arbitrary lengths (Line 106). It would be great to quantify this by generating stories of varying lengths and validating whether there is a loss in quality after a certain length.
- One particularly interesting component of the method is the Visual Context Module, so I would have liked to see far more evaluations performed on it. For example, the authors mention that multiple frames can be used for conditioning by concatenating their CLIP feature. Some interesting questions that could help strengthen the importance of the component include:
    - How much was this evaluated?
    - How much does conditioning on more frames assist in temporal consistency?
    - How many previous frames can be concatenated without hindering performance?
- Stating that the model achieves a significant improvement in the alternative models seems like a strong over-claiming when approximately 30 participants were used for the user study. A substantially larger pool of participants would be needed to truly quantify this improvement, especially since this is the only relevant metric used to evaluate the methods. Why are the reported FID metrics between Table 1 and Table 2 different? Is a different dataset used? I would expect only “without HF” to be different if both use the same dataset.


**Questions:**

I hope the authors can help clarify several questions that I have regarding both the method and the evaluations.

**General Comments:**
- A reference to Rombach et al. is missing when discussing Stable Diffusion (e.g., in Line 38 where it is first used).
- First stated contribution is that the authors propose the task of open-ended visual storytelling, but this was a previously studied problem (as also mentioned in Line 44).
- When describing the style transfer module and visual context module, the authors reuse the same notation for the weight matrices \(W^K\) and \(W^V\). To make the writing a bit clearer, the authors should use a different notation for each case since these are separate modules if I understood correctly.

**Method:**
- What happens if the image conditioning is simply done using the standard image-to-image technique where the noised latent is conditioned on the previous image? Adding an ablation study would help validate this design choice. If performing a thorough ablation study is difficult, providing intuition on the motivation for the Visual Context Module would help highlight the contribution to readers.
- Where is the style transfer model illustrated in Figure 2? Is this the blocked labeled LoRA? The authors mention that this is a LoRA-like architecture. Could the authors kindly clarify what exactly the difference between the original LoRA design is and why this modification was made?
- Regarding the Multi-Frame Conditioning:
    - According to Figure 2, it seems like StoryGen receives several previous frames as conditioning while Line 136 indicates that only the previous frame is used for conditioning. Could the authors kindly clarify this?
    - Similarly, in the general setting, it seems that multiple frames are used for conditioning. Therefore, would it be more accurate to revise the equation in Line 150 to indicate any number of \ell frames as conditioning?
    - In Section 5.1, the authors mention that the multiple-frame fine-tuning stage is done using a single conditioning image (Lines 256-257). Where do the multiple frames come into play? Is this really a multi-frame conditioning?

**Small comments:**
- Line 70: ou -> our
- Line 142: generate -> generates, align -> aligns
- Line 276: Propmt -> Prompt


**Limitations:**

The authors include a discussion on current limitations and potential societal impacts in the supplementary.

---

> ### Author Rebuttal · Authors · 2023-08-10
>
>   Thanks for the positive feedback on our task setting, architecture and results. Hope the response below will resolve your confusion and thus raise the score accordingly. We are always open to further discussion.
> - W1&2. The name and necessity of human feedback
>   - Please refer to Q5 of the global response. We consider the manual filtering process as a reflection of human preference, and a larger amount of human feedback data will lead to further improvement in performance.
> - W3. Lack of comparison with StoryGAN, Story-DALL-E, and AR-LDM
>   - Please refer to Q6 of the global response. They are indeed weaker baselines compared with SDM.
> - W4. FID metrics or CLIP-based metrics
>   - Please refer to Q8 of the global response. We provide evaluation results on CLIP score and PickScore(CLIP-based) in the rebuttal PDF.
> - W5-1. Ablation studies on BERT-like masking
>   - Please refer to Q7 of the global response. We provide quantitative and qualitative results in Table 2 and Figure 5(rebuttal PDF).
> - W5-2. Ablation studies on directly fine-tuning on the story narrative
>   - We provide quantitative results on FID and CLIP-based scores in the rebuttal PDF. As shown in Table 2(rebuttal PDF), the model fine-tuned with story narrative has demonstrated a noticeable decline in performance, which agrees with our hypothesis that text aligning closely with the prompt style of SDM will yield superior image quality.
> - W6. Evaluation on generating stories of varying lengths
>   - We provide a qualitative experiment result in Figure 3(rebuttal PDF). We agree with the reviewer, in order to train the model to generate longer image sequences, more training data and computation are required.
> - W7. Ablation studies on visual context module
>   - The effectiveness of Visual Context Module has been demonstrated in the supplementary by comparing StoryGen-Single with StoryGen, quantitatively in Table 1 and qualitatively in Figure 2,3,4 (supplementary).
>   - We present the FID and CLIP-based results on StorySalon test set of StoryGen with different numbers of conditioned frames in Table 3(rebuttal PDF). More condition frames will lead to quantitative improvement, but not very significantly. This also agrees with our convenient inference setting of using a single conditioned frame.
> - W8-1. A larger pool of participants
>   - Thanks, we will be more cautious with the tone, and find more participants in the final version. In future work, we aim to construct an online platform, that enables to collect human feedback on a much larger scale.
> - W8-2. Results of Table 1 and Table 2
>   - Please refer to Q3 of the global response.
> - Q1. Reference on L038
>   - Thanks, we have cited this paper in L031.
> - Q2. The novelty of open-ended visual storytelling
>   - Please refer to Q1 of the global response.
> - Q3. The notation abuse
>   - Thanks! We will add additional footnotes for the classification of the projection matrices in the style transfer module and visual context module.
> - Q4. Condition the noisy latent on the previous image
>   - Thanks for the suggestion, we have indeed tried some other methods including DDIM inversion and concatenating the noisy latent with the previous image, but none of them worked well. The intuition is that these methods tend to keep the layout of the image unchanged and the content(i.e. character appearance) different, which is against our problem scenario that the content is unchanged and the layout different. However, we agree that they are interesting directions to be explored in future work.
> - Q5. Details of LoRA-like architecture
>   - To augment the expressiveness of LoRA, we use a larger intermediate representation dimension in its hourglass-like architecture. Due to this misalignment with the requirement of low rank, we refer to this module as LoRA-like architecture.
> - Q6-1&6-2. Inconsistency between Figure 2 and L136, L150
>   - Please refer to Q4 of the global response.
>   - Figure 2 is to demonstrate the ability of our model to condition on multiple frames, and L136 and L150 describe the practical setting for convenience. We have added new experiments on multiple conditioned frames in Table 3(rebuttal PDF), which shows insignificant performance improvement.
> - Q6-3. The name of multiple-frame fine-tuning stage
>   - Multiple-frame fine-tuning is relative to single-frame pre-training. We have added new experiments by conditioning on multiple frames, we will clarify this in our final paper.
> - Q7. Small comments
>   - Thanks! We will correct the typos in the final paper.

---

### Official Review · Reviewer_WrUp · 2023-07-02

**Soundness:** 3 good
**Presentation:** 4 excellent
**Contribution:** 2 fair
**Rating:** 6
**Confidence:** 5

**Summary:**

This work proposes the model StoryGen for the task of visual storytelling. Visual storytelling is a task to generate a sequence of consistent images given a story (several sentences). StoryGen is a diffusion model taking in both image and text as conditions, and outputs an image consistent with the conditions. The training process includes pre-training for single image, finetuning for multiple image and finetuning with human feedback. On top of the StoryGen model, this work also provides a dataset, called StorySalon, which consists of 2k story books (30k well aligned text-image pairs).

The overall structure of StoryGen model is simple. It is built upon existing well trained diffusion models and image/text encoders. To generate cartoon-like image, LoRA is adopted into the text conditioning module in a diffusion model. The author calls it the style-transfoer module. The parameters in LoRA are updated at this pre-training stage to give single cartoon image. Next is the multipe image fune-tuning. StoryGen conditions on both text and image, which is implemented by using two cross attention layers: One is noise input + text and the other is noise input + encoded previous generated image. After the second step, StoryGen if further finetuned on 100 high-quality stories.

The author also spends some efforts to collect the StorySalon dataset. To begin with, the author downloads a huge number of videos and subtitles from online web resources with potential stories. Then give story-level description and visual level description for each story. The story level description is obtained by using dynamic time warping algorithm using subtitles. The visual level description is derived from ChatCaptioner. Finally, OCR method is applied to get potential videos captions.

The experiment section shows that StoryGen model can give consistent and story-like output images, while other methods fail.


**Strengths:**

The paper is clearly written. The collected StorySalon dataset could benefit the research community.

**Weaknesses:**

There is not much technical novelty, and the experimental results are limited.

**Questions:**

- Line 192 mentions the 100 high quality books are added into the training set.  For the third stage training, is it still using the datasets from previous steps, but with 100 more samples? Would it make more sense to only use the 100 high quality books for further fine-tuning? Also, the name (fine-tuning with human feedback) is confusing since there is neither human feedback nor reward model. Is there any human involved to give preferences of different generated outputs from the StoryGen model?

- The human feedback fine-tuning stage doesn’t seem to help a lot in the quantitative scores. It would be better to include visual examples without human feedback fine-tuning. If the difference is too small, it would be better to re-structure the paper.

- For the visual examples given in Figure 4, I noticed the boy with yellow hair shows in both stories. Does this boy appear a lot in the training set? Also, it would be better if the author could provide more than one examples for each story using StoryGen. Otherwise, it feels like the model is overfitting to the training data.

**Limitations:**

- Since there is no human labeler involved in collecting StorySalon dataset and the total number of stories in StorySalon is only 2k, I’m concerned about its quality.

- It is also unclear from the examples provided if the results is derived by just overfitting the training set.

---

> ### Author Rebuttal · Authors · 2023-08-10
>
> Thanks for your affirmation and appreciation for our writing and dataset. Hope the response below will resolve your confusion and thus raise the score accordingly. We are always open to further discussion.
> - W1. Not much technical novelty
>   - Please refer to Q1 of the global response.
> - W2. Limited experimental results
>   - We have presented additional ablation studies on further Human Feedback and Visual Context Module in Table 1(supplementary). These experiments prove that further Human Feedback fine-tuning with more data will lead to a better FID score, and Visual Context Module will also benefit the model. During the rebuttal period, we provide more qualitative and quantitative results in the rebuttal PDF.
> - Q1-1. The use of 100 books
>   - In Human Feedback process, we are using the original dataset along with the 100 generated new storybooks, which contain around 600 more text-image pairs. In Table 1(supplementary), we scaled human feedback data to 700 stories and around 3k pairs, and achieved further improvement in terms of FID compared to fine-tuning with 100 more storybooks.
>   - Compared with our whole StorySalon dataset, the improved human feedback data of 3k samples still appears to be a relatively small amount of data. Directly fine-tuning StoryGen with these storybooks may cause catastrophic forgetting and destroy the pre-trained text-image alignment in the latent space of SDM.
> - Q1-2. The name of human feedback
>   - Please refer to Q5 of the global response. We consider the manual filtering process as a reflection of human preference.
> - Q2. The effects and visualization of human feedback
>   - Please refer to Q5 of the global response. A larger amount of human feedback data will lead to further quantitative improvement. And Figure 2,3,4(supplementary) show that the model with augmented human feedback fine-tuning(3k samples) could produce images with better consistency and quality. So human feedback is effective, but requires a larger screening data to significantly reflect its value.
>   - Besides, as presented in L189-L190, another potential advantage of Human Feedback is to avoid potentially scary, toxic or biased content, which can not be measured by quantitative metrics.
> - Q3. The yellow-haired boy and overfitting
>   - Thanks for pointing out this. After checking the training set, a yellow-haired boy with a similar looking does appear in the data, though not frequently. However, we disagree with the comments that this refers to overfiting, as shown in recent work[1], diffusion models can potentially generate samples similar to seen images, regardless of the scale of the training set.
>   - Besides, as shown in Figure 2,3,4(supplementary), the visual results of StoryGen-Single (no Visual Context Module) show explicit inconsistency compared with StoryGen and StoryGen-HF, which also proves that we achieve consistency in generation by Visual Context Module instead of overfitting.
>   - [1] Carlini et al. EXTRACTING TRAINING DATA FROM DIFFUSION MODELS.
> - L1. The StorySalon dataset quality
>   - There are human labellers involved in collecting the StorySalon dataset, which has been mentioned in L215-L216. In the data collecting process, we also thoroughly went through the dataset multiple times, manually checked and removed the frames that do not satisfy our demand but were not found by automatic filtering.
>   - Please refer to Q1 of the global response for our data quality. Our StorySalon has a similar data scale on image-caption pairs compared with previous datasets, and surpasses them with a much larger vocabulary and a far longer average story length.
> - L2. Overfitting
>   - Please refer to Q3 in this response. More qualitative results are presented in Figure 1(rebuttal PDF) for demonstrating the diverse generation ability of our model.

---

> > ### Comment · Reviewer_WrUp · 2023-08-21
> >
> > Thanks for the author's response. This is an interesting work. I'll keep my score as weak accept.

---

### Official Review · Reviewer_tcbg · 2023-07-03

**Soundness:** 3 good
**Presentation:** 3 good
**Contribution:** 2 fair
**Rating:** 3
**Confidence:** 4

**Summary:**

The work focuses on the application of image generation based on a given story. Specifically, the proposed model is conditioned on the current sentence and prior generated images to ensure the story is engaging and coherent. A progressive training strategy is proposed to achieve a good model. To improve the proposed method, a new dataset is collected, while a set of human-verified generative samples are also utilized to improve the generated images.

--------------------------
I acknowledge the author's effort in the rebuttal and have made changes to the review accordingly.

**Strengths:**

+ This work demonstrates the possibility of generating visual storytelling images conditioned on the given stories.
+ A three-stage curriculum training strategy is proposed to train the proposed model. However, it would be great to demonstrate the limitation of training the model with multiple-frame (i.e., without single-frame pre-training)
+ The authors collected a large-scale dataset to enable model training for storytelling purposes.

**Weaknesses:**

- the technical contribution of this work is limited. Most of the components are not novel and the key contributions are the way it is combined to generate a plausible output. It is unclear what the insights generated from this work that is not previously obvious to the community.
- The description of the new StorySalon dataset is limited. Specifically, it is unclear if the collected dataset has obtained legal consensus and properly handled copyright issues.
- The work lacks a comparison with existing work, such as those introduced in line 44 and line 93-102. The two baselines in Table 1 are too naive as both are inherently limited to generate a fair comparison with the proposed method.


Minor:
- Fig 2 should clearly state that the Image encoder only considers a single previous frame to generate the next frame.

**Questions:**

- Please justify what it means by "open-ended". Was the dataset divided in a manner that the training and validation/test dataset has distinct distribution or content?

- The final stage of curriculum training strategy fine-tune the model with human feedback. Here, the stories are generated with ChatGPT and the images are synthesized with the same model. This creates a problem of bootstrapping machine learning and makes it unclear whether such an approach can benefit the model. A recent work [1] discussed the curse of recursion with generated data. I want the invite the author to discuss the problem of this.

In addition, the results in Table 2 show that with or without HF has a small difference. Can the author also explain what does 0.19 translate to the differences in the qualitative results?

[1] Shumailov et al. THE CURSE OF RECURSION: TRAINING ON GENERATED DATA MAKES MODELS FORGET.  https://arxiv.org/pdf/2305.17493.pdf

- In line 141, why is a small number of newly-added trainable parameters required and how does it impact the model performance/learnability?

- In Table 1 and Two, why is the FID score in both table inconsistent?

**Limitations:**

The paper (in supplementary) discusses that data bias is an issue that needs to address in this domain. Collecting a larger dataset for training is the solution discussed. This may be valid considering this work is still in the early stage of the research.

I want to point out that the discussed approach is limited as (1) it is resource-consuming, and (2) it will face the problem of copyright in order to obtain a good dataset for training. The data ownership issue may be a major hindrance.

---

> ### Author Rebuttal · Authors · 2023-08-10
>
> Thanks for your affirmation and appreciation of our writing and dataset. Hope the response below will resolve your confusion and thus raise our score accordingly. We are always open to further discussion.
> - S2. Limitation of training the model without single-frame pre-training
>   - We have stated in the manuscript of our submission (L152-L158 and L177-L182) that the Style Transfer Module trained via single-frame pre-training allows StoryGen to quickly adapt to the style of storybooks. Without this stage, it would be difficult to generate the first frame with the correct style. There is an obvious domain gap between the style of the real-world image and that of the training data for Visual Context Module, which will finally weaken the effect of multi-frame finetuning.
> - W1. Limited technical contribution
>   - We disagree, please refer to Q1 of the global response.
>   - Besides, our insights also lie in thinking about the limitations of existing generative models and how to extend them to new tasks, specifically how to collect appropriate data and design suitable architectures.
> - W2. Limited description of the StorySalon dataset and copyright issues
>   - Please refer to Q1 and Q2 of the global response. We have presented a visualization of a portion of data in Figure 3, and we have also performed statistics on the categories included in StorySalon, as shown in Figure 1(supplementary), confirming the diversity of our data. More examples can be seen in Figure 2(rebuttal PDF).
> - W3. Baselines and comparison with existing work
>   - Please refer to Q6 of the global response. Previous works are even weaker baselines compared with SDM and Prompt-SDM, so we do not compare with them. Besides, we compare our model with StoryGen-Single shown in Table 1(supplementary) and fine-tuned SDM in Table 2(rebuttal PDF), which are both stronger baselines.
> - W4. Illustration of Figure 2
>   - Please refer to Q4 of the global response.
> - Q1. The meaning of 'open-ended'
>   - Please refer to Q1 of the global response. More detailedly, the meaning of 'open-ended' can be interpreted from two different perspectives:
>   - On one hand, for previous Story Visualization and Story Continuation tasks, StoryDALL-E or AR-LDM tend to overfit datasets containing only a few characters, such as FlintstoneSV and PororoSV, and can not generalize to other datasets or prompts, that is, closed or occlusive.
>   - On the other hand, SDM is pre-trained on large-scale text-image pair data, and can generate novel images with novel text prompts. Our StoryGen model inherits the prior knowledge of SDM and thus can extend to generate novel story frames with novel storylines.
>   - Benefiting from StorySalon dataset and StoryGen model, we can prompt ChatGPT to generate a series of new storylines, and our model can generate new visual stories with new characters, not limited to only a few protagonists as in the past. This is why our new task is called 'open-ended visual storytelling'.
> - Q2. The problem of bootstrapping and the small performance difference with or without HF
>   - Please refer to Q5 of the global response. A larger amount of human feedback data does lead to further improvement in quantitative performance. As for the bootstrapping problem, this has been shown effective in InstructGPT and ChatGPT, thus we follow similar training routines as those work.
> - Q3. The necessity of the newly-added trainable parameters and the impact
>   - As stated in L140-L141, our StoryGen model can be regarded as a semi-frozen SDM with a small number of newly-added trainable parameters, which specifically refer to our proposed Style Transfer Module and Visual Context Module. The former aligns the style of images generated by StoryGen to storybook images, and the latter enables our model to use the generated previous frame as a visual condition for autoregressively generating a coherent story sequence.
>   - Results in Table 1, and Figure 2,3,4 of the supplementary can illustrate the effectiveness of our proposed modules both quantitatively and qualitatively.
> - Q4. Inconsistent FID score
>   - Please refer to Q3 of the global response.
> - L1. Discussion on resource and copyright problems for a larger dataset
>   - Please refer to Q2 of the global response. The data processing pipeline contains multiple carefully designed steps to ensure the quality of data, but each step does not need to consume many computational resources. We will open-source our data processing pipeline to the community. Once researchers have suitable and feasible data sources, they can easily expand our StorySalon dataset.

---

> > ### Comment · Reviewer_tcbg · 2023-08-16
> >
> > Thank you for the details respond to the review and additional results in the provided PDF.
> >
> > Can the author provide additional comments about the video data of the proposed dataset. While "we will release in the form of YouTube URLs" is a plausible practice, it has been evidenced that video could be removed from the platform and unaccessible by other researcher. This could potentially affect the reproducibility and availability to other researchers.
> >
> > About the bootstrapping problem, I hope the authors can provide more discussion based on the findings in [1]. Do also analyse the generated feedback and training size. The amount of data in ChatGPT is on different scale when compared to the dataset presented in this submission. It is hard to determine if the bootstrapping effect is positive or negative for this work.

---

> > > ### Author Response · Authors · 2023-08-18
> > >
> > > - Thanks for your comments.
> > > - Releasing datasets in the form of URLs is a common practise, for example, Youtube-8M [2], WebVid-10M [3] and VideoCC [4]. However, we agree with the reviewer that the videos can be removed from the website, thus we are currently expanding our StorySalon dataset, and will replace all YouTube video data with open-source ebook data registered under **CC BY 4.0 license** in order to completely eliminate copyright concerns.
> > > - Thanks for pointing out the paper, we have read through it in detail. As mentioned in the response, in our case, we do observe positive gains by doing human feedback, as shown in **Table 1 (supplementary)**. This might be due to the fact that paper [1] has only investigated MNIST, which is a significantly simpler dataset than the ones we are using.
> > > - In addition, we do observe that these gains are correlated with the volume of feedback data, our current human feedback data for fine-tuning only constitutes a mere **10%** of the StorySalon dataset, which is also significantly smaller in comparison to the LAION-5B employed in SDM pre-training. Thus we are actively developing a user platform to gather more user feedback, aiming to explore the positive improvement upper bound of human feedback on our StoryGen model and analyze whether a negative effect would emerge as human feedback expands to a certain scale.
> > > - [1] Shumailov et al. THE CURSE OF RECURSION: TRAINING ON GENERATED DATA MAKES MODELS FORGET
> > > - [2] Sami Abu-El-Haija et al. YOUTUBE-8M: A LARGE-SCALE VIDEO CLASSIFICATION BENCHMARK
> > > - [3] Max Bain et al. FROZEN IN TIME: A JOINT VIDEO AND IMAGE ENCODER FOR END-TO-END RETRIEVAL
> > > - [4] Arsha Nagrani et al. LEARNING AUDIO VIDEO MODALITIES FROM IMAGE CAPTIONS

---

### Official Review · Reviewer_SemY · 2023-07-05

**Soundness:** 2 fair
**Presentation:** 3 good
**Contribution:** 3 good
**Rating:** 3
**Confidence:** 4

**Summary:**

This paper presents StoryGen, an auto-regressive image generator that leverages text and image conditioning. StoryGen incorporates a style transfer module integrated into the text-conditioning module, along with a visual context module. The authors also constructed a substantial dataset called StorySalon, comprising 2K storybooks and 30K text-image pairs.


**Strengths:**

1. This paper constructs a new dataset StorySalon contains 2K storybooks and more than 30K well-aligned text-image pairs. The authors have invested significant effort into filtering the data, making it a valuable resource for advancing the field of story visualization.
2. The paper is well-written and easy to follow.

**Weaknesses:**

1. The illustration does not align with the description provided. In line 135, the authors state that "StoryGen generates the current frame $\mathcal{I}_k$ by conditioning on both the current text description $\mathcal{T}_k$ and the previous frame $\mathcal{I}_{k-1}$, as illustrated in Figure 2." However, the left figure of Figure 2 shows the image conditioned on more than one previous image, which contradicts the mentioned conditioning approach.
2. The improvement of Human feedback appears to be trivial, as indicated in Table 2. The 0.19 FID score gap could potentially be attributed to different training seeds, which raises doubts about the significance of the reported improvement. (I do not agree with the statement that 200 stories are too small since the model is trained using 2k stories overall. It appears to be sufficient for human alignment and does not require an extensive amount of data.)
3. The FID score lacks precision in the test set, particularly with only 100 storylines. It is recommended that the authors expand the test set by including more stories to provide a more accurate evaluation.
4. The baselines SDM and Prompt-SDM are too weak.  It is suggested that the authors compare StoryGen with finetuned or LoRA-finetuned SDM models using the same training settings to establish a more robust baseline for comparison.
5. The auto-regressive generation approach employed by StoryGen has already been proposed by AR-LDM. Consequently, the architecture design itself lacks novelty.
6. StoryGen is only conditioned on one previous image and does not utilize the corresponding caption of the previous image. In the depicted cases of Figure 1 and Figure 4, there is only one main recurring character. If multiple characters were present, StoryGen may struggle to ground the characters in the previous images. Furthermore, if a character does not exist in the previous image, StoryGen may face difficulties in maintaining consistency between frames.
7. The language understanding capacity of StoryGen appears to be weak. For instance, in the second case of Figure 4, the rabbit appears small in the fourth frame, whereas it should be as big as it is in the fifth frame. Additionally, in the sixth frame, the boy's hair does not become brighter as described in the caption. Moreover, in the seventh frame, multiple other boys are depicted with the same yellow hair, which contradicts the previous story setting. This limitation may stem from StoryGen solely relying on the only one previous frame $\mathcal{I}_{k-1}$ and not incorporating previous captions into its generation process.

**Questions:**

1. StoryGen generates the current frame $\mathcal{I}_k$ by conditioning on current text description $\mathcal{T}_k$ and the previous frame $\mathcal{I}_{k-1}$.  I am wondering why the authors do not use all historical frames $\mathcal{I}_0, \cdots, \mathcal{I}_{k-1}$ in this process.
2. StoryGen introduces a style transfer module $\phi_{text}$ and a single-frame pre-training stage. However, since the training set is constructed from multiple sources (as mentioned by the authors in Line 244), the style is inconsistent. Moreover, many LoRA style transfer plugins not only tune the cross-attn module but also the self-attn modules. Why not follow this setting and add LoRA layers in both self- and cross-attn layers? This way, StoryGen can also benefit from other style transfer LoRA developed by the community.
3. In Line 186, the authors mention randomly dropping some words in the text with a certain probability following BERT. I have doubts about this technique because there is no masked language modeling task in StoryGen, and such a technique may not be helpful.
4. I question whether the authors truly need to replace descriptive captions with story narrative text. The story in the storybook and the story generated by LLMs are both in the form of story narrative text, rather than descriptive captions. Using story narrative text to train the model may lead to implicit alignment with human performance.

**Limitations:**

1. The author should provide examples of the constructed dataset showcasing different visual styles and character appearances.
2. The StorySalon dataset consists of 2K storybooks and over 30K well-aligned text-image pairs, which is smaller compared to datasets such as FlintstonesSV (24K stories and 123K image-caption pairs), PororoSV (14K stories and 74K image-caption pairs), and VIST (27K stories and 136K image-caption pairs). Despite the authors' claim that StoryGen can perform open-ended story generation, it remains unclear whether StoryGen can generate stories involving more complex scenarios with unusual entities.
3. Apart from human feedback, the authors have not conducted any other ablation studies to evaluate the effectiveness of their proposed techniques, such as word dropout and curriculum learning.
4. There are concerns regarding the legality of using web-crawled e-books. The authors should provide additional information about the sources of the e-books and clarify whether proper copyright guidelines were followed.

---

> ### Author Rebuttal · Authors · 2023-08-10
>
> Thanks for your affirmation on our writing and dataset. Hope the response will resolve your confusion and raise our rating accordingly. We are always open to further discussion.
> - W1. Inconsistent illustration
>   - Please check Q4 of global response.
> - W2. Limited improvement of human feedback
>   - Please check Q5 of global response.
> - W3. The precision of the FID score
>   - Please check Q3 of global response. FID on StorySalon test set can be considered to accurately reflect the distribution of generated results.
> - W4. Stronger baselines
>   - As presented in Q1 and Q6 of global response, we choose SDM and Prompt-SDM as baselines, as no existing model owns better open-ended generation ability. We also provide results of fine-tuning cross-attn layers of SDM on StorySalon in Table 2(rebuttal PDF) and fine-tuning SDM with LoRA in Table 1(supplementary). StoryGen performs significantly better than them.
> - W5. Novelty of auto-regressive generation
>   - Please check Q1 of global response.
>   - We did cite AR-LDM in our work. As far as we know, it is an arxiv paper, thus should be treated as concurrent work. Moreover, despite both approaches consider autoregressive generation, there are critical differences between StoryGen and AR-LDM:
>   - (i) AR-LDM uses the heavy BLIP model and integrates image information into text embedding, while StoryGen proves that CLIP image encoder can be directly used with an independent module to attend to image condition; (ii) AR-LDM has to train all the parameters end-to-end including LDM, CLIP and BLIP, while StoryGen favours lightweight training with far fewer parameters to optimise.
> - W6. Previous captions and character problem
>   - We agree. Our current model may potentially suffer from multiple characters/character loss.
>   - We have tried adding previous captions but this leads to the ignoring of image condition. We have further investigated the idea of multiple condition images, as shown in Table 3(rebuttal PDF). However, the performance gain is limited. We conjecture that this is due to our dataset prior, as we are training/evaluating the model on children's books, that naturally prefer simple stories with a single protagonist. This is an interesting question for our future research while generalising towards sequences of complex stories and natural images.
> - W7. Weak language understanding ability
>   - We agree. This is because StoryGen adopts pre-trained SDM, that uses CLIP text encoder, which is known to suffer several limitations, for example, it can not distinguish complex spatial or quantitative relationships, and struggles with affirmative and negative. This challenge can be alleviated by stronger text-to-image models like Deep IF, which is regarded as future work.
> - Q1. Conditioning on multiple frames
>   - Please check Q4 of the global response. We present FID and CLIP-based results on StorySalon test set of StoryGen with multiple condition frames in Table 3(rebuttal PDF). The performance gain is not significant, possibly due to our dataset prior towards simple stories with a single protagonist, which agrees with our convenient choice to use a single conditioned frame during inference. We will also include the multi-frame results in our revised paper.
> - Q2. Data style inconsistency and LoRA design
>   - The data style inconsistency is trivial compared with the strong expressive capacity of LoRA.
>   - Here, we aim to learn the style of storybooks with as few parameters as possible, potentially maintaining the original prior of SDM to the greatest extent. It is a cool idea to insert different LoRAs into the model, but we always need to fine-tune the Visual Context Module to maximize its ability when using LoRA with large style differences. So different LoRA designs have no difference in training.
> - Q3. BERT masking
>   - Please check Q7 of global response.
> - Q4. The choice between descriptive captions and story narrative text
>   - We provide quantitative results on FID and CLIP-based scores in Table 2(rebuttal PDF). The model trained with story narrative shows worse performance, which agrees with our hypothesis that texts aligning closely with the prompt style of SDM yield superior image quality. Therefore, during inference, we will first convert narrative texts into descriptive captions with LLM to maximize the generative ability of the model.
> - L1. Examples of StorySalon
>   - We present a visualization of a portion of data in Figure 3, and we also show statistics on object categories in StorySalon in Figure 1(supplementary), confirming the diversity of our data. More examples can be seen in Figure 2(rebuttal PDF).
> - L2. The scale and diversity of StorySalon, and open-ended capability
>   - Please check Q1 of global response. Although other datasets contain more stories, the amounts of image-text pairs are actually in the same scale as ours, because these datasets use a fixed story length (5 frames) while stories in our dataset are far longer compared to theirs. Besides, our dataset contains the largest vocabulary till now, thus it is clearly more diverse than these datasets.
>   - The 'open-ended' capability is from two sources: (i) our StoryGen inherits the open-set prior of SDM and can ideally generate image sequences of objects, scenes of arbitrary category, i.e., an extension of image-based generative model; (ii) our storylines generated by ChatGPT can be varing drastically, beyond those seen in our training data. The diversity of our generated results is visualized in Figure 1(rebuttal PDF).
> - L3. More ablation studies of proposed modules
>   - We have provided further ablation study of our proposed modules and curriculum training in Table 1(supplementary). The stepwise performance improvements show the effectiveness of our Visual Context Module and curriculum training including human feedback. More ablation results on word dropout are presented in Table 2 and Figure 5(rebuttal PDF).
> - L4. Concerns about the legality of e-books
>   - Please check Q2 of global response.

---

> > ### Comment · Reviewer_SemY · 2023-08-19
> > **Thanks for the author response**
> >
> > Thanks a lot for the response. I would like to keep my score as it is.

---

### Official Review · Reviewer_4bmh · 2023-07-07

**Soundness:** 3 good
**Presentation:** 3 good
**Contribution:** 3 good
**Rating:** 5
**Confidence:** 4

**Summary:**

The paper propose an approach for fine-tuning diffusion models for the task of story generation, where a model must generate frames for sentences in a story. To do so, they propose adding adaptors conditioned on both images and text into a pre-trained stable diffusion UNet. The authors also introduce a scraped dataset of 2k stories with 30k image-text pairs, which serves as the data foundation for their fine-tuning.

**Strengths:**

S1. The dataset of story text and images is a significant contribution, which can be useful for future work on visual storytelling.

S2. The paper is generally well framed and motivated. Writing and presentation is in general strong and polished.

S3. Simplicity of the method. The method uses off-the-shelf components and algorithms (LoRA, cross-attention, etc.) to enable new capabilities. I see this simplicity as a strength not a weakness.

S4. Inclusion of human evaluation is a strength.

**Weaknesses:**

W1. Presentation. Figure 2 suggests that StoryGen is conditioned on all past frames. However, in reality StoryGen is conditioned on the most recent frame.

W2. Results. It appears that the model is not able to preserve style and content as well as say DreamBooth. For example the stripes on the shirt in Figure 4 are not preserved.

W3. Evaluation. It seems that the StoryGen model without human feedback model was not evaluated in the human evaluation (Table 1). Given that the ablation in Table 2 shows similar FID scores for StoryGen with and without human feedback, it is not clear if this step is really necessary.

W4. Lack of baselines. StoryGen is specifically fine-tuned for the desired task, while stable diffusion is not. Given this, it is perhaps not surprising hat StoryGen greatly outperforms stable diffusion. Can some other baselines, perhaps based on the DreamBooth (with the subject in the first image representing the special [V] token) be used for a stronger baseline? This is just one idea; however, comparing against some prior work may help to elucidate the strength of the method.

W5. Evaluation. Fine-tuning can sometimes hurt the generality of a model. How well are individual frames generated relative to the base SD model? Is it possible to do a human evaluation here or compute CLIP scores? Ideally, the story would be cohesive (frame-to-frame consistency) without degradation in quality of each frame.

**Questions:**

Q1. Can the authors provide further discussion of why StoryGen without human feedback was not evaluated in the human evaluation (W3)? Is it possible to add this evaluation? I believe this would help contextualize the importance of the human feedback in the proposed method.

Q2. Can the other concerns related to evaluation be addressed (W2, W4, W5)?

**Limitations:**

The paper does not directly address limitations in the main paper, which is an additional weakness. I suggest discussing failure cases or otherwise conducting a failure analysis.

---

> ### Author Rebuttal · Authors · 2023-08-10
>
> Thanks for the positive feedback on our writing, dataset and simplicity of the proposed method. We hope the following response can fully resolve the raised concern and thus raise our score accordingly. We are always open to further discussion.
> - W1. Presentation
>   - Please refer to Q4 of the global response.
> - W2. Comparison with DreamBooth
>   - We agree that DreamBooth may perform better. However, DreamBooth is an optimization-based method that has to finetune the **entire** model for many iterations(\~20 minutes) on each reference image sequence (3\~5 images). While our method is learning-based, once trained, it can be efficiently applied to generate coherent stories without any special fine-tuning.
>   - In fact, the idea of DreamBooth can also be integrated into our framework to enhance the ability to maintain the style and content via test-time adaptation. We treat this as future work.
> - W3. Evaluation on human feedback
>   - Please refer to Q3 of the global response for details of these two Tables.
>   - Considering that human evaluation for each model variant is very labour-intensive and time-consuming, we can only afford to do it on the experiments for comparison with state-of-the-art methods. While for ablation study on validating the effectiveness of our proposed module, we adopt the widely used FID metric, the experiment results can be seen in Table 1 of the supplementary material.
>   - Please refer to Q5 of the global response on the analysis of further human feedback.
> - W4. Lack of baselines:
>   - As presented in Q1 and Q6 of the global response, there is no existing model with the capabilities for open-ended visual storytelling, so we choose SDM and Prompt-SDM as two strong baselines.
>   - DreamBooth has two insurmountable limitations for our task: First, it requires thousands of steps(\~20 minutes) of adaptation for each reference image sequence (3\~5 images), which is very time-consuming and difficult to extend to large-scale test data. Second, the input of visual storytelling only contains a storyline from users or ChatGPT without reference images, while DreamBooth requires 3\~5 reference images for optimization. Thus it can not be finetuned to be a baseline.
> - W5. Evaluation on individual frames:
>   - Thanks for the advice. In fact, we have included the experiments for single-frame finetuned model, i.e., StoryGen without Visual Context Module in Table 1(supplementary), termed as StoryGen-Single. The FID score of StoryGen-Single is significantly better than that of SDM and Prompt-SDM. Figure 2,3,4(supplementary) also demonstrate that StoryGen-Single can generate story-style images closer to the given storyline than SDM and Prompt-SDM.
>   - Please refer to Q8 of the global response for CLIP-based metric evaluation results.
> - Q1. Effectiveness of human feedback:
>   - Please refer to W3 of this response and Q5 of the global response. Besides, as presented in L189-L190, another potential advantage of Human Feedback is to avoid potentially scary, toxic or biased content, which can not be measured by quantitative metrics.
> - Q2. Evaluation of performance:
>   - As discussed above, we have provided quantitative and qualitative experimental results in the supplementary that further illustrate the effectiveness of our proposed method, demonstrating that our single-frame pretraining, multi-frame finetuning and human feedback gradually improve the model's performance. For more evaluation results, please check the rebuttal PDF.
> - L1. Limitations and failure cases:
>   - Due to the limited space of the manuscript, we have included the limitations in supplementary (L114-L124) and provided the visualization of failure cases in Figure 4(rebuttal PDF). The first two cases show that StoryGen will produce low-quality characters when the character number in the image is large and the spatial relationship is complex. The 3rd case shows that CLIP text encoder is not good at counting(wrong number of objects, CLIP mistakes 3 hedgehogs to 2) and the last case shows the bias of our dataset towards page-like images with a crease in the middle.

---

> > ### Comment · Reviewer_4bmh · 2023-08-14
> >
> > Thanks for the responses to my comments! However, I do still feel that an additional baseline that has some sort of task-specific training is necessary to contextualize performance. DreamBooth was one such idea, but other ideas could include LoRA fine-tuning a StableDiffusion model on the proposed StorySalon dataset. I am electing to keep my initial score of 5.

---

> > > ### Author Response · Authors · 2023-08-15
> > >
> > > - Thanks for your comments. But the results of additional baselines that you required have been provided in our supplementary and rebuttal. Please check **Q6** of the **Author Rebuttal** again.
> > > - The results of StableDiffusion with LoRA fine-tuning on StorySalon have been provided in **Table 1** of the **supplementary**, termed as **StoryGen-Single**. Besides, we have also included the StableDiffusion model with cross-attn layers finetuned on our proposed StorySalon dataset as another strong baseline, namely **Fine-tuned SDM**, in **Table 2** of the **rebuttal PDF appendix**. We compared our model with these stronger baselines on FID and CLIP-based scores, and the quantitative results have demonstrated that our full StoryGen can also significantly outperform these stronger baselines.

---

### Author Rebuttal · Authors · 2023-08-10

We appreciate all reviewers for the valuable comments and feedback. Hope the following response can fully resolve the raised concerns. We will release all codes, datasets, and models for future research purposes.
- Q1. Novelty and Contribution(ALL)

  We would like to start the rebuttal by elaborating our contributions in this paper:
  - (i) **On proposed novel task**. Unlike previous Story Visualization or Story Continuation performed under limited characters/vocabulary, we consider a more challenging yet exciting task, i.e. _open-ended visual storytelling_, that requires generative models to generate **coherent** story frames based on storylines given by users or LLMs under **open-ended** vocabulary, e.g. free-form storylines and novel characters. The diversity of our generated results is shown in Figure 1(rebuttal PDF).
  - (ii) **On our dataset and processing pipeline**. Previous FlintstoneSV and PororoSV datasets only contain 7~9 characters with limited vocabulary and story length, incompetent for open-set task. We design a complete data processing pipeline, screen data suitable for our task from videos and ebooks, and build **StorySalon** dataset with a quite large vocabulary, which contains thousands of characters from hundreds of categories.
  - (iii) **On architecture design and training scheme**. We aim to inherit powerful **open-set** text-to-image generation capability of pre-trained StableDiffusion. Benefiting from our designed Style Transfer Module and Visual Context Module, **StoryGen** can generate coherent story-style image sequences with given storylines by simply training a small number of parameters. Surpassing strong baselines, StoryGen proves the effectiveness of proposed modules.
Overall, we present our preliminary efforts on initiating research on open-ended visual storytelling that has lots of potential room for further improvement.
- Q2. Potential copyright issues(SemY&tcbg)
  - For video data in our StorySalon dataset, we will release in the form of YouTube URLs; and for ebook data, we obtain ebooks from global digital libraries registered under **CC BY 4.0 license**, e.g. Bloom Library. All books are open-source. The community can process raw data via our processing pipeline.
- Q3. FID in Table 1 and Table 2(SemY&tcbg&zaiF)
  - Apologies for the confusion. Due to limited space, the explanation of experiment settings was included in L031-L071(supplementary), we will clarify this in the revised paper.
  - Concretely, in Table 1, we prompt ChatGPT to generate 100 storylines and perform visual storytelling conditioned on the storylines. FID score is calculated between the generated 100 visual stories (~600 frames) and the test set ground truth of StorySalon;
  - In Table 2, we use text of StorySalon test set as condition to generate image sequences. FID score is calculated between the synthesized test set (~200 stories, ~3K frames) and the ground truth test set.
- Q4. Illustration of Figure 2 on #frames(4bmh&SemY&tcbg&zaiF)
  - Apologize for the confusion. Our proposed architecture is indeed flexible on conditioning multiple frames, which can be achieved by concatenating their CLIP features as visual contexts, as mentioned in L165-L166.
  - During rebuttal, we tested our model on more condition frames. As shown by FID results of StoryGen with multiple condition frames in Table 3(rebuttal PDF), more contexts do improve performance but not very significantly, we conjecture this is due to our dataset/task prior, i.e. simple stories for children. We will add the results and discussion to our final paper.
- Q5. Effectiveness of human feedback(ALL)
  - Our Human Feedback process consists of two stages: Prompt ChatGPT to generate storylines, and manually filter out high-quality images from our generated samples, by simply giving YES or NO labels. This manual filtering procedure is thus a reflection of human preference.
  - We agree with the reviewers that adding small-scale human feedback seems to be not effective, and this is the reason for the limited gain brought by human feedback in Table 2.
  - In Table 1(supplementary), we expand the amount of data and obtain StoryGen-HF. It further improves performance compared to StoryGen-hf, suggesting that human feedback is effective, though it requires a larger scale annotation to reflect its value. Visual examples in Figure 2,3,4(supplementary) also show that human feedback can help to produce images with improved consistency and quality.
- Q6. Baselines(ALL)
  - Considering the **open-ended** feature of our task, SDM is the only strong baseline. **StoryGAN** and **StoryDALL-E** naturally do not support open-ended task due to the lack of large-scale pre-training and outdated backbones, it is thus unfair to compare with their released checkpoints. **AR-LDM** has not provided pre-trained checkpoints, and as the model is trained end-to-end, our computation resource and dataset scale are insufficient for training it.
  - In Table 1(supplementary), we have provided an additional baseline **StoryGen-Single**, denoting SDM finetuned on StorySalon with Style Transfer Module. And StoryGen performs significantly better in FID. Please check the rebuttal PDF for more baseline results. Qualitative results in Figure 2,3,4(supplementary) also demonstrate that StoryGen outperforms StoryGen-Single in terms of inter-frame coherence, showing the effectiveness of Visual Context Module.
- Q7. BERT masking(SemY&zaiF)
  - We treat this as empirical discovery, rather than technical contribution. We provide our experimental results during project developments, as included in the rebuttal PDF.
- Q8. CLIP-based metric(4bmh&zaiF)
  - Thanks for the advice, we provide evaluation results on CLIP score and PickScore(CLIP-based) in the rebuttal PDF. But since CLIP is trained on real-world image-text pairs, these metrics explicitly prefer natural images, as shown by low scores of GT(cartoon style). So CLIP-based metrics can only be treated as a reference.

---

### Author Response · Authors · 2023-08-19
**Kind Reminder**

Dear Reviewers,

We would appreciate it if you could share with us whether our rebuttal has addressed your concerns.

We would also be happy to answer if you have any further questions.

Thank you very much.

---

### Decision · Program_Chairs · 2023-09-21

**Decision:**

Reject

**Comment:**

Overall, all reviewers agreed that this work addresses an interesting problem. However, nontrivial concerns about novelty and evaluation remained to be published. By reflecting reviewer’s initial and post-rebuttal comments, this submission would be much for future submission.